# Diving into Real-World Practicum in Physical Education: Deconstructing and Re-Signifying Pre-Service Teachers' Reflections

Eugénia Azevedo [1,*], Ana Ramos [1], Carla Valério [2], Rui Araújo [3] and Isabel Mesquita [1]

[1] Centre of Research, Education, Innovation and Intervention in Sport (CIFI2D), Faculty of Sport, University of Porto, 4200-450 Porto, Portugal; agramos@fade.up.pt (A.R.); imesquita@fade.up.pt (I.M.)
[2] Faculty of Education, Southern Cross University, Bilinga, QLD 4225, Australia; carla.valerio@scu.edu.au
[3] Research Centre in Sports Sciences, Health Sciences and Human Development, University of Maia, Avenida Carlos de Oliveira Campos—Castêlo da Maia, 4475-690 Maia, Portugal; ruiaraujo@umaia.pt
* Correspondence: up201001611@edu.fade.up.pt

**Abstract:** Reflection is crucial for aspiring teachers, yet many pre-service teachers (PSTs) struggle to grasp its true meaning. This study explores how PSTs deconstruct their initial reflections and re-signify their understanding based on challenges encountered in real-world practicum settings. Additionally, it examines how the facilitator supported the PSTs' reflections over time. Over the course of a year in the physical education teacher education context, six PSTs, along with the first author, who fulfilled dual roles as external facilitator and researcher, engaged in three iterative Action Research (AR) cycles. Within each AR cycle, the external facilitator explored the authentic teaching challenges faced by PSTs, offering individualized support while unpacking reflection concepts. Data were collected through focus group interviews, reflective journals from the PSTs, and the observation of participants by the external facilitator, which provided contextual field notes on the PSTs' teaching–learning experiences. Our findings emphasize the need to initially understand PSTs' views on reflection. This serves as a starting point for deconstructing the three concepts outlined in our theoretical framework, through the scrutiny of PSTs' real teaching–learning experiences. This process facilitated a resignification, leading to an advanced comprehension of reflection among the PSTs. The study emphasizes the value of integrating this approach into systemic teacher education reforms and suggests extending training and mentorship to cooperating teachers.

**Keywords:** reflective practice; physical education teacher education; facilitator role

## 1. Introduction

Nowadays, reflection is considered a crucial skill for pre-service teachers (PSTs) who need to be 'thoughtful and alert' during their educational journey to become effective practitioners [1–3]. Accordingly, most teacher preparation programs advocate for reflective practice to help PSTs navigate the challenges of the teaching–learning process and improve their efficacy and proficiency [4]. Despite these encouragements, many PSTs continue to struggle with both practicing and understanding reflection. This struggle is heightened in Physical Education Teacher Education (PETE), where subject intricacies and a myriad of complex, non-linear, and diverse challenges further complicate reflection [5,6].

Overall, integrating reflection into teaching practice empowers PSTs to navigate teaching challenges, interpret practical experiences, and identify areas for growth in uncertain circumstances [7]. Indeed, active participation in the process of reflection nurtures the cultivation of reflective practice, characterized by heightened self-awareness [8]. Through reflective practice, PSTs can become empowered, enhancing professional growth and profound transformation [9]. Nevertheless, the reflective practice process presents challenges and complexities that hinder inexperienced teachers from fully engaging in this transformative process. Consequently, when asked to reflect on their teaching–learning experiences,

PSTs tend to provide summaries with descriptive rather than analytical insights [10,11]. This challenge primarily arises from their tendency to prioritize content knowledge in teaching–learning practice [12], resulting in a surface-level understanding of reflection as a routine task in their profession, lacking in-depth interpretation [13]. Another fundamental issue that has been consistently highlighted in the literature involves the challenges that PSTs often face due to their limited scope in reflective practice. These challenges often arise from PSTs' unclear understanding of how to engage meaningfully in reflection [14]. Such limitations not only restrict the type of reflective practice, but also often fail to consider critical factors within the teaching–learning experiential context [15]. This, in turn, limits the overall reflective growth of PSTs.

Therefore, it is of the utmost importance for PSTs to grasp the nuances of their reflective practice, including the ability to differentiate between superficial and deep reflection [16] and to understand how to immerse themselves in a practice that embodies critical reflection. To achieve this, the PSTs must be capable of deconstructing their preconceived ideas about reflection, enabling them to take the first step towards a critical evaluation of their initial understandings. Indeed, PSTs need to develop the ability to actively participate in critical reflection and interpret their own experiences [14]. However, PSTs are often unaware of their reflection level and the specific topics they are considering. This requires a constructivist approach (where learners construct their own knowledge connected to their prior experience and knowledge embedded in meaningful contexts) [17] that includes both the content and process of reflection. In terms of content (e.g., examining the PSTs' reflections on their teaching practice, such as planning, classroom interactions, and faced challenges), it provides insights into how these experiences shape their reflective practice. Regarding the process (e.g., understanding the nature of reflection, identifying its practical contexts, clarifying goals, and defining how to nurture it), it involves a deeper exploration of the reflective practice itself. A vital component of this constructivist approach is the resignification phase, wherein PSTs reconstruct their conceptions of reflection. This phase involves adopting a more profound and analytical viewpoint, extending beyond the simple act of reflection. It encompasses their engagement in interpreting and critically applying their reflective insights within the teaching and learning contexts. In doing so, PSTs will be better prepared to navigate the challenges inherent in their experiences effectively and progress towards becoming adept critical thinkers [18]. While a substantial body of empirical research in PETE has explored various strategies and learning environments to foster reflection and critical reflection [19,20], there remains a noticeable gap in the literature regarding how PSTs effectively internalize the concepts and processes of reflection, as well as the difficulties they encounter in recognizing the personal effort that reflection demands. One critical factor influencing the internalization of the reflective process among individuals is learning that it is contextualized, situated within relevant contexts, and directly tied to their teaching–learning experiences, thus fostering a sense of inclusion [21]. Indeed, research has highlighted that the value of PSTs' reflections may be tied to the sociocultural context in which they learn and in which reflection is enacted [22,23]. The practicum provides an immersive opportunity to enhance PSTs' critical reflection by interpreting their challenges and extensively comprehending their teaching–learning experiences [9]. Thus, integrating the teaching–learning experiences gained during practicum with reflective practices can serve as a valuable strategy for guiding PSTs in addressing the specific challenges they are likely to face throughout their careers as educators [24]. This immersive endeavor enables PSTs to self-regulate their learning effectively [25] and cultivate a multi-perspective thinking ability [26]. Moreover, it empowers them to critically scrutinize the widely accepted practices in physical education (PE) [19], which, in turn, facilitates the development of PSTs' critical reflection [9].

While empirical evidence underscores the advantages of reflective practice during the practicum [27–29], much of the prior research has been concentrated on the theoretical-practical nexus within this context [30,31] or on PSTs' perceptions regarding the content of reflection [32]. This highlights a pressing need to investigate how these theoretical

foundations are seamlessly integrated into PSTs' daily experiences, facilitating their ability to address challenges and refine their teaching–learning process. Moreover, despite this recognition, particularly in PETE, many PSTs struggle to reflect during practicum due to the perception of having 'too many things going on' [28]. This challenge underscores the need for more contextualized and experientially relevant approaches to help PSTs internalize the reflective process and tackle the concrete challenges they will face in their professional journeys. Among the key factors facilitating the development of meaningful and productive reflection in PSTs is the necessity for continuous instruction, guidance, and deliberate practice involving both supervisors and peers [33]. However, existing research on PETE has notably fallen short of exploring this crucial type of support. For instance, Vogelsang, et al. [34] point out that in practicum, reflection is not sufficiently supported for PSTs. Creating a supportive environment, providing ongoing instruction, and facilitating collaborative reflection are crucial for PSTs to meaningfully engage in the reflective process [35]. Through meaningful dialogue and active listening, facilitators (i.e., supervisors) can empower PSTs to navigate the complexities of their practice [36] and foster critical decision making [33]. Therefore, facilitators can guide PSTs to deconstruct reflection [37] and explore experiences from diverse perspectives [38]. Furthermore, there is a lack of interventional research exploration that enables, stimulates, and explores the intentional influence of a facilitator on scaffolding the development of PSTs' reflection process through the interpretation of their practical problems. An Action Research (AR) design can address this need by fostering ongoing improvement through iterative processes and personalized support (e.g., guidance from a supervisor or facilitator), empowering PSTs to enhance their understanding and application of reflective practices within the practicum [39]. Accordingly, adopting an AR design, the aim of this study was to explore how PSTs deconstruct their initial reflections and subsequently re-signify their understanding, based on challenges encountered in real-world practicum settings. Additionally, it examines how the facilitator supported the PSTs' reflections over time.

## 2. Theoretical Framework—Reflection Concepts

*Reflection, Reflective Practice, and Critical Reflection*

In this study, understanding the dimensions of reflection—reflection, reflective practice, and critical reflection—as non-hierarchical stages and interconnected concepts that emerged from analyzing the practical difficulties of PSTs is crucial to offering distinct yet interrelated conceptualizations, contributing to a comprehensive interpretation.

Hence, the term 'reflection' in this study is used in two interrelated contexts. Firstly, it is a cognitive process used to navigate complex problems [40], involving synthesizing ideas, questioning narratives, and integrating knowledge to better understand personal subjectivity and experiences [41]. This iterative process requires a sustained re-examination of ideas and an active and conscious intention to comprehend experiences and generate fresh insights [42]. Secondly, 'reflection' also denotes the initial stage in the continuum of developing reflective abilities, where PSTs begin to engage with the concept at a fundamental level. In parallel, reflective practice encompasses the practical application of the reflective process in daily decision making and problem solving [43]. It builds upon the foundational stage of reflection, extending it into the realm of practical teaching scenarios. Critical reflection represents a further deepening and broadening of this process. It is a complex transition influenced by multiple factors, such as diverse perspectives, varying historical backgrounds, and socio-political contexts [44]. In addition, it involves a comprehensive understanding that extends beyond the individual's personal experiences to include broader societal and institutional dynamics. This level of reflection includes reflecting on established assumptions about teaching, learning, self, school, and society, as well as education's societal and political impacts; this entails the ongoing analysis, questioning, and critique of these assumptions, leading to transformative changes in practice [18].

The process of deconstruction–resignification, which is integral to this framework, illustrates how PSTs revise and redefine their understanding of these reflection dimensions.

Deconstruction refers to critically examining and questioning the existing practices and understandings, particularly in response to the challenges encountered in their practicum. This process aligns with the constructivist principles of learning through experience and reflection. As PSTs deconstruct, they are actively engaging with their experiences, reflecting upon and questioning their prior understandings and practices.

Resignification occurs as PSTs integrate the new insights gained from this critical examination. This leads to a redefined and enriched approach to reflection that is more directly aligned with their practical experiences. In a constructivist approach, this process is fundamental as it allows learners to reconstruct their knowledge and understanding based on their experiences, fostering deeper and more meaningful learning. Thus, the journey of deconstruction and resignification not only enhances their conceptual understanding, but also brings theory and practice into a closer, more harmonious relationship.

## 3. Methodology

### 3.1. Study Design

A qualitative approach was employed in this study to investigate not only the PSTs' understanding of reflection, but also how the facilitator supported their reflections during the practicum. This approach is suitable for exploring practical themes and understanding the 'who, what, and where' of events or experiences [45]. Situated within the interpretative paradigm, the study acknowledges reality's diverse and subjective nature and knowledge's socially constructed nature, embracing ontological relativism and epistemological constructionism [46].

This study followed an AR design [47], with an interventionist approach [48] across three cycles conducted during a school year (from September 2016 to June 2017). Each AR-cycle corresponded to a distinct school year period. The first author, acting as the External Facilitator (EF), had a multifaceted role focused on enhancing PSTs' identification of areas for improvement, promoting inclusive participation, and empowering those PSTs who lacked confidence. Through processes of thinking, planning, acting, evaluating, and rethinking, the EF aimed to understand PSTs' practice issues and provide individualized support, incorporating the deconstruction of reflection concepts. This process stemmed from the genuine challenges faced by PSTs in their teaching practicum. To facilitate this process, the first AR cycle included a diagnostic phase. This phase was introduced to allow the EF to become familiar with the PSTs' understanding of reflection, thereby guiding subsequent interventions to address the real needs and challenges they faced during their practicum [49]. This phase also helped the EF choose pertinent content for subsequent AR cycles and scaffold PSTs' deconstruction of theoretical concepts.

In line with the fundamental tenet of AR, which emphasizes collaboration and the improvement of actions [50], PSTs were active participants in the research process; they were not merely objects of study, but co-constructors of their own learning [51]. This involvement primarily included examining and interpreting their own experiences during their practicum.

### 3.2. Context and Participants

The research took place in a two-year PETE Program and focused on six PSTs in the second year of their master's program (i.e., practicum). This particular PETE program included a reflection component during the practicum, specifically by requiring PSTs to keep reflective journals. These PSTs were supervised by a cooperating teacher from the host school and a supervisor from the university. The practicum was conducted in various schools, with this particular group of PSTs being assigned to three different schools. Participants were selected based on availability, willingness to participate and school placements [52]. The EF, holding a master's degree in PE and being a PhD student in Sports Science, was well suited to the study's interventionist approach, given their deep academic engagement with the subject. All participants were informed of the scope of the research and their right to withdraw from the study. Consent forms were signed by all

participants, and anonymity was established by using pseudonyms. The host university's ethical commission approved all the procedures.

### *3.3. Data Collection*

The data collection sequence started with focus group interviews (FG), followed by participant observations (PO), and concluded with Reflective Journals (RJ). This specific order was designed for a holistic understanding of the PSTs' reflection. The initial focus groups helped identify key themes and questions that were further explored during participant observations. Finally, Reflective Journals allowed the PSTs to self-reflect on their experiences, thereby enriching and triangulating the data gathered from the earlier methods. The continuous flow of the AR process, alongside its inherent phases, was complemented by the interplay between these three data collection methods.

### *3.4. Focus Group*

The reflective semi-structured FG discussions were conducted by the EF [53,54] for two reasons: first, to explore PSTs' understanding and thoughts on their practicum experiences, aimed at fostering their reflective capacity and improving teaching; second, to facilitate discussions on key concepts related to the process of reflection such as reflective practice and critical reflection. The EF cultivated a social environment of trust [55], promoting the dialogue and critical thinking essential for reflective growth. The EF prompted PSTs into reflection by discussing and interpreting their teaching challenges. These three FGs were evenly distributed throughout the academic year, capturing early, mid, and late stages, held in faculty settings, audio-recorded, lasting approximately 110 min each, and transcribed verbatim.

### *3.5. Participant Observation*

The EF conducted PO intending to explore the 'how' and 'why' behind the PSTs' practical experiences [56]. This entailed observing the PSTs' spontaneous actions within their teaching environment, allowing the EF to interpret the PSTs' teaching practices and generate a more comprehensive understanding of the practicum experience. Through these observations, the EF gained insight into the consistency between the PSTs' actions and their previously expressed ideas, and insight into how well they applied the reflective content discussed in the previous FG. This information guided the formulation of questions for the next FG session, which aimed to help the PSTs relate to their experiences and deconstruct the theoretical concepts they had learned [57].

### *3.6. Reflective Journal*

The RJs were assembled by the PSTs as part of their activities, but also were intentionally structured and scaffolded by the EF to deepen their understanding of reflective concepts. Through writing, the PSTs could explore and critically analyze the challenges encountered in practice, leading to a more profound comprehension of reflection [58]. Writing in the journals facilitated the systematization of their ideas regarding their practical experiences and the reflective concepts discussed in the previous FG session. Furthermore, following each PO, the EF suggested additional avenues for exploration, focusing on specific problems observed that required further clarification. This contributed to a deeper understanding of the PSTs' teaching experiences and connected their practical observations with the reflective concepts discussed in the FGs.

### *3.7. Data Analysis and Research Procedures*

In AR, analysis continues to be conducted throughout the three cycles and the overall analysis of the research [51]; this is to reflect the on-the-spot, interactive and cyclical epistemological nature of AR [59]. The research followed a deductive process, using theory to identify what the PSTs knew about reflection, which in turn guided the formulation of research aims, interview questions, and further theoretical content.

Collaboration played a central role in this research, involving six PSTs engaged in practicum and the EF. Through collaboration, the aim was to deepen and expedite the understanding of the ongoing action, leveraging the diverse experiences of the PSTs to foster personal and professional growth [60]. This objective was achieved by facilitating group discussions, providing theoretical frameworks for debate (grounded in discussions, participant observations, and PSTs' written reflections), and scaffolding PSTs in articulating their perspectives and experiences.

The EF played a crucial role in the process of deconstructing complex reflection concepts, breaking them down into more understandable parts. This was achieved by providing a framework and guidance to clarify and challenge the PSTs' preconceptions, leading to a deeper understanding of reflection. In contrast, the PSTs deconstructed their teaching–learning practices and initial reflections. This self-reflective process involved critically examining and re-evaluating their experiences and practices in light of the theoretical knowledge provided, fostering a more nuanced and comprehensive understanding of reflection.

In this context, the focus of deconstruction and resignification varied. The EF's deconstruction was geared towards clarifying reflection concepts, while the PSTs focused on critically analyzing their teaching methods and experiences. The process of resignification occurred as the PSTs integrated this new understanding into their teaching practice, leading to a refined and enhanced approach to teaching and learning. This approach not only aligned with the analysis and exploration of specific issues during the PSTs' practicum [61], but also completed a cyclical journey of understanding and applying reflection concepts in practice (see Figure 1).

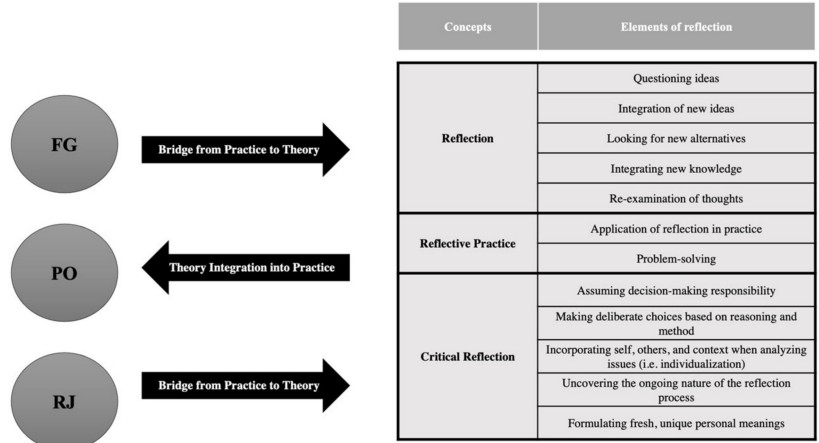

| Concepts | Elements of reflection |
|---|---|
| Reflection | Questioning ideas |
| | Integration of new ideas |
| | Looking for new alternatives |
| | Integrating new knowledge |
| | Re-examination of thoughts |
| Reflective Practice | Application of reflection in practice |
| | Problem-solving |
| Critical Reflection | Assuming decision-making responsibility |
| | Making deliberate choices based on reasoning and method |
| | Incorporating self, others, and context when analyzing issues (i.e. individualization) |
| | Uncovering the ongoing nature of the reflection process |
| | Formulating fresh, unique personal meanings |

**Figure 1.** Interplay of reflection concepts and data collection methods.

The data underwent a thematic analysis [62] process to analyze and interpret the information. In the initial stage, the first author was immersed in the transcribed data, reading them multiple times to become familiar with their contents. Relevant segments of data were identified and labelled with initial codes. This coding process was systematic and rigorous, aiming to establish 'a solid foundation' for developing themes [59]. The first author engaged in discussions with the last author to refine the codes and identify themes. Through collaborative dialogue, the last author offered insights that unveiled new subtleties in the data, prompting the first author to revisit and refine the coding and themes meticulously. The themes were then moved from a surface level, based on the explicit content shared by the PSTs, to a deeper level, focusing on the underlying ideas conveyed by the PSTs [63]. The next phase involved generating initial themes relevant to the research question, addressing concepts and organizing the codes into coherent themes. The data were reviewed repeatedly in the fourth phase to ensure the analysis accurately represented the data and aligned with the research question. The fifth phase involved a thorough review of the data by the first author to prevent misrepresentation through coding and content

structuring. The analysis was repeatedly checked to ensure alignment with the research framework and maintain data integrity. Finally, in the sixth phase, the identified themes were (re)named to reflect the temporal impact of the AR's constructivist epistemology. The process involved writing the report, fully analyzing developed themes, and presenting a concise, coherent, and logically structured data account. These main themes are presented in the 'Results' section, providing insights and findings from the analysis.

*3.8. Trustworthiness*

Several criteria were implemented to ensure trustworthiness in the study [64]. Firstly, participants in the FG were encouraged to express their honest opinions and perspectives without being influenced by others, with each individual's thoughts valued equally [65]. Secondly, data triangulation was employed by utilizing multiple data collection sources, such as FG, PO, peer observations, and RJ, to gain different perspectives on the phenomena under investigation [66]. Thirdly, regular peer briefings and data analysis within the research team were conducted to mitigate the influence of subjective perspectives and promote the rigorous scrutiny of the data. Lastly, the PSTs were frequently probed about the underlying meanings of their actions and verbal expressions, while also acknowledging disciplined subjectivity through the recognition of the researcher's own subjectivity [67]. These procedures aimed to regulate the subjective aspects inherent in qualitative research and enhance the overall trustworthiness of the findings [53].

## 4. Results

*4.1. The 1st AR-Cycle—Reflection Understood as a Container of Ideas or a Checking System—'I Understand Reflection as the Organization of the Elements Included in the Lesson Plan and the Realization of What I Did or Didn't Do after the Lesson'*

In the first FG, the facilitator tried to understand how the PSTs understood reflection and its application to teaching–learning practice. The EF aimed to identify the PSTs' baseline to develop further nuanced reflective abilities: 'Facilitator—do you reflect on your practice? What does reflecting mean to you?' How does reflecting help you when you are teaching? Give me an example'. The PSTs clarified that the fact it was a required task during the practicum meant that it was naturally performed: '*Maria*—yes, it's one of the many things that are imposed on us throughout the year, so we reflect'. However, the PSTs understood reflection as merely recounting their actions in the teaching–learning process. They simply provided a narrative of what had transpired in their most recent class without engaging in any critical thinking: '*António*—I analyze the material I'm going to cover, for example, and how I'm going to plan it and then how I managed to execute it in class'.

The awareness of applying reflection in practice for the PSTs was related to the act of describing and reporting all the elements integrated into the lesson plan: '*Carlos*—I go to class, try to review the entire plan in my head to see if I remember everything I had planned, try to see how many students I'll have, the materials. . . those things, and of course it always makes me very anxious'.

Also, during the PO, the facilitator noted that the PSTs engaged in limited analysis, ignoring other factors that could contribute to the lesson's success: 'Facilitator—aren't there other important things you should be aware of? *Maria*—the important things are the ones that are here (lesson plan)'. On the other hand, the facilitator understood the limitations of the PSTs in their inability to establish relationships between what was planned and what should be done: 'Facilitator—*Maria* realized that the class did not understand the exercise and still did not intervene'. In parallel, PSTs struggled to apply reflection in practice, as they observed their teaching but failed to interpret it effectively: '*Facilitator*—Paulo observed that the students were playing around and did not realize the need to change the class dynamic'.

In written reflection, PSTs focused on the execution of pre-established tasks. They exhibited a thought based on reporting, describing, and verifying events that occurred during the teaching–learning practice: '*António*—it was raining, so the lesson had to be

taught on the stairs (. . .) all students integrated into the groups and showed attention during the instruction phase'.

Furthermore, the PSTs presented a rumination of ideas, that is, the ideas were repeated and adhered to the possibilities integrated into the lesson plan, not analyzing the reasons that justified changing the planned exercises: '*Gustavo*—I thought it was important to change this exercise organization (. . .) I modified the learning situation according to the alternatives I had included in the plan (. . .)'. Similarly, when PSTs demonstrated the ability to integrate new elements into their analysis, such as student behavior, they limited themselves to constantly applying the same strategy. PSTs could not reflect on their actions, in the sense of trying to understand them: '*Paulo*—this behavior of the students cannot happen again' (. . .) whenever this behavior arose, I reprimanded them'.

*4.2. The 2nd AR-Cycle—Reflection Understood as a Means to Discover One's Own Doubts and Interpret Practice—'I Reflect to Open My Mind Beyond the Obvious'*

In the previous cycle, the PSTs were not aware of how limited their reflection was and tended to associate it with the idea of 'depositing ideas'. So, to help the PSTs understand their limitations regarding what reflection truly means, as well as how limited their understanding of reflection was and how it could be applied (i.e., reflective practice), the facilitator introduced and explained the concepts of reflection and reflective practice. Here, the facilitator intended to desconstruct the idea of what reflection is, regardless of the levels of reflection, so that they could understand what reflection and practical reflection are. The EF used concrete examples from practice to engage the PSTs and help them identify with the process: '*Facilitator*—(. . .) reflection involves understanding the reasons for our actions, questioning the decision, evaluating its impact, and trying to understand what we can do to improve as professionals (. . .) Reflective practice is related to the interpretation of your own practice'.

The deconstruction of reflection and reflective practice by the facilitator, through specific examples from the PSTs' practice, led to the PSTs becoming aware that their way of thinking did not include reflection: '*Maria*—wait, so we weren't reflecting, but just stating?' and that if they were actually to reflect, they could be more effective in their teaching: '*Gustavo*—if I think, or rather reflect, in that way, it could help me improve my lessons'.

Based on their renewed understanding of both concepts, the PSTs were able to raise questions, specifically about the relevance of integrating new ideas into their reflections, '*Gustavo*—wouldn't it be productive to think about other things? Or because we are PSTs, do we always need to think in a PSTs way?', as well as the need to get more involved and take responsibility for decision-making, '*Maria*—wouldn't it also be important, for example, to have a stronger attitude, to impose ourselves more when we disagree with you CT or I don't know, to step forward and act?'. Throughout the dialogue, PSTs also started to look for new alternatives: '*Paulo*—then it would be important to consider other options instead of always applying the same solution'.

However, during the facilitator's explanations, the PSTs showed that they did not understand the differences between reflection and reflective practice: '*Gustavo*—OK, but now I don't know if I get it, are they the same [reflection and reflective practice]? They seem synonymous'. For this reason, the facilitator provided a clarification about the concepts of reflection: '*Facilitator*—well then, reflection is more related to a more introspective, more intimate action, applying a set of skills such as questioning, analyzing, evaluating, confronting to make a decision (. . .) is more closely related to a process of questioning your actions and thoughts. Through this questioning process, you will begin to incorporate and examine new ideas'.

Then, the facilitator deconstructed the meaning of reflective practice, referring to situations that had occurred in the teaching–learning practice of the PST in question, to create more explicit definitions of the concepts: '*Facilitator*—in your activity as a PST (. . .) you will interpret why that solution applied in your classes did not work and look for new solutions (. . .)'. When listing some reasons for the need for its implementation, 'because

this way you have more abilities not to be surprised or swallowed up by problems, you are dependent on yourselves and not on others' opinions or what you have seen done (…)'. This explanation leads PSTs to recognize the need for actively solving their problems (i.e., problem-solving) during teaching instead of ignoring what is happening: '*Carlos*—I see, so we must think about what we can do differently in our classroom. It's not just about following the lesson plan, but also about being alert to the students' responses and adapting our methods accordingly'.

At this stage, PSTs already demonstrated the ability to analyze and change their actions, but they had difficulty positioning themselves at the center of the reflective process. The facilitator introduced and clarified the concept of critical reflection to help PSTs understand that they were a fundamental piece of the puzzle in terms of including themselves in the reflection process. For this purpose, the facilitator helped them understand that doubt was one of the key elements in developing critical reflection: '*Facilitator*—implies that you take a central position, you are the owners of your decisions (…) In other words, it is more associated with your analysis, where you assume, for example, that negative states can distort your analysis of ideas. By being aware of it, you prepare not to be carried away by the heat of the moment'.

They acknowledged that although some of these questions may have arisen before, they were reluctant to express them because they did not usually include them in their analyses of lessons. *António* said the following: 'This was something we hadn't even thought of, and even if we had, I was afraid to look ignorant by asking about it because these questions are not part of our lesson analyses'. This allowed them to begin to understand and recognize the importance of including elements of critical reflection, such as multiple perspectives, in their analyses. *António* continued with the following: 'Only now have I realized I should explore other options, relativize, and not just focus on the problem or that particular lesson'.

During the dialogue, the facilitator aimed to support PSTs in developing decision-making skills and their ability to make deliberate choices. She drew on a situation from the PSTs' teaching–learning practice to develop a characteristic element of critical reflection (i.e., focusing the reflection on themselves): 'So, *Carlos*, what do you think about how you present yourself when you teach? Have you ever considered how this might influence your students? For example, in the dance class, how did the way you interacted with them, your disposition, and your tone of voice affect your class and their response?'. By shifting the focus of their reflection to themselves, the PSTs realized they could generate new interpretations of the same situation. *Carlos* said the following: 'Oh, wait, if I start looking at what I am like, me, *Carlos*, in the class? Then I already have other perspectives, I really do… Now I have to think about how I will talk to them'.

By assuming decision making and making deliberate choices, the PSTs developed new understandings, specifically regarding how their lack of enthusiasm could affect the students and limit their involvement in the lessons. *Carlos* added the following: 'Now that I think about it, I may have ignored my own lack of motivation when teaching these dance classes, and that could have affected the students' motivation'.

Similarly, in the RJ, the PSTs demonstrated the ability to analyze possible reasons for what happened in order to have more options for the future. *Maria* stated: 'The students were constantly stopping their activities, and I have to think about why this happened'. She also reflected on what they should do to change, incorporating their own analysis in a self-critical process: 'I know that in the next class since they don't like the subjects that day, I'll have to think of a class that they'll enjoy more, that's more fun and gives them more time for activities, with fewer moments of instruction. Basically, make the class more appealing because then I know they will participate and engage much more'.

After the PO, the facilitator noted a need for the PSTs to analyze some elements related to critical reflection (i.e., their own actions and the students) and adapt their actions accordingly. *Gustavo* asked *Sara* the following: 'In class, were you thinking about the consequences of your yelling? You were a bit robotic; you didn't think that the little group

was only responding that way because you confronted them head-on. It was a matter of 'wanting to win your way to the end,' you forgot to ask those little questions.'. Specifically, the PSTs expressed difficulty in applying critical reflection during teaching and became aware of what they could not do when faced with a problem. *Sara* admitted the following: 'Yes, it was stronger than me; when they provoke me, I think of nothing else, I just want to complete the task. . . so yes, you're right'.

*4.3. The 3rd AR-Cycle—Reflection Understood as a Process That Generates Self-Knowledge: 'When I Reflect, I Delve into the Reasons for My Actions, Seeking to Understand Them'*

Although the PSTs had revealed an understanding of critical reflection as a process of making choices from doubts that involved taking responsibility for their actions, in the third cycle, the facilitator identified a need for further development in this area, specifically concerning self-knowledge. To address this, the facilitator deconstructed the remaining elements of critical reflection through various strategies.

Initially, the facilitator encouraged the PSTs to interpret the problems faced by their colleagues, aiming to develop their ability to act in the face of a problem (i.e., incorporating others when analyzing issues): 'Facilitator—I suggest that you analyze each other's practice, as some have already been doing it in class'. Thus, the PSTs demonstrated that when facing their colleagues' problems, they became aware of the importance of adopting various strategies to guide their practice: '*Gustavo*—Look, I thought about this after that class. . . if she [colleague] had considered different scenarios she might have been more prepared; she often seems to act on the spot without anticipating, but anticipation is something we're all learning to do, that's what we're taught (. . .)'.

As the discussion progressed, the facilitator deconstructed another element of critical reflection, namely the ability to relate events and intentionally attribute meaning to their actions, using a specific moment (i.e., incorporating self when analyzing issues): 'Facilitator—Your students had already questioned you about your solution, in dividing the groups yourself, was there any reason to impose your decision? Did it have to do with any aspect of the class? And why did you do it?'. The PSTs showed the discovery of new solutions and the establishment of relationships between them: '*António*—The question is really knowing how to defend it and now I know (. . .)'. They also developed their own meanings by integrating the 'how' and 'why' behind their actions: 'I can explain why I cared more about those students and not others (. . .)'.

To deepen the development of meanings, the facilitator asked the PSTs for an explanation related to the new elements (i.e., their own actions and the students) that they had mentioned in the previous RJs: 'Facilitator—And in what sense can these new elements help you in your practice?'. The PSTs demonstrated that they had learned to commit and be accountable to themselves for the problems and their attitudes (i.e., they attributed meanings to their actions), which allowed them to increase control and security in the process: '*Gustavo*—Look, I don't blame everyone and everything now, I know that above all, I am the one steering the ship, so I assume my decisions (. . .)'.

Similarly, in the RJs conducted in this AR cycle, the PSTs expressed a change in their attitude to an idea that was now more complete and interacted with various elements. This was not only with themselves and the students, but also with the context in which they were inserted (i.e., incorporating self, others and context when analyzing issues): '*Sara*—(. . .) I realized, however, that my thinking is very dependent on the students; everything was around them, and now I see a broader picture (. . .) there are them, but there's also me and the circumstances'.

Concurrently, the PSTs recognized the 'relative world' in which they found themselves (i.e., practicum) as a place where nothing could be taken for granted and unquestionable. It is, in fact, an ongoing and nonlinear process, and therefore the importance of adapting to the context is emphasized: '(. . .) Gustavo—The issue here is being able to give different answers even to the same problems (. . .)' of the complex and dynamic nature of the reflective process '(. . .) because if we just change a comma in the situation and that solution may no longer be

good'. The PSTs realized that there were no unique solutions in the reflective process, but that constant learning was required: '*Maria*—This is very difficult. I found myself trying a thousand and one approaches, but they didn't work. It's not enough to know the formula and apply it, there are no unique solutions, it's constant work (...)'. They also realized the depth that finding an appropriate solution could sometimes require: '(...) I now know that this needs work and dedication, it's not just about knowing the theory, that's the starting point'.

Thereafter, the facilitator turned to an event observed in the PSTs' teaching–learning practice to develop their exploration of unique personal meanings, exploring the notion of the individualization and adaptation of actions according to the specificities of each student: '*Facilitator*—So why did you choose to constantly change the way you spoke with them? For example, in one of the groups, did you give much less autonomy than in the last class?'. In response, the PSTs expressed an understanding of the importance of continually reflecting on their practice, as this was the only way they could realize what needed to be changed: '*Sara*—This is a long process that needs to be done all the time, so now I focus on adapting my interaction to each student and not generalizing'. They also conveyed a sense of certainty that their concerns would be ongoing, rather than transient or limited to the practicum period, reflecting a broader intention for professional growth (i.e., awareness of the value of self-analysis in professional development): '*Gustavo*—I currently find that my concerns will never disappear but will be different from these (...) only with this attitude can I become a good teacher'.

During the PO, the PSTs revealed an awareness of the importance of reflection, the ability to identify their mistakes and find solutions: '*António*—I think I didn't explain that well. I should have adapted the exercise for this group that wasn't getting it'. In addition, the PSTs demonstrated an integration of new perspectives, specifically those of the students; they valued what the students thought, and thus were genuinely receptive to their suggestions: '*Facilitator*—At the end of the class, *Gustavo* asked the students how they would evaluate him, requesting suggestions for improvement'.

Despite making progress in understanding the concepts of reflection, reflective practice, and critical reflection, the EF realized that some PSTs still faced difficulties in incorporating their 'selves' when analyzing issues: '*Facilitator*—Maria still struggles to understand how she should distance herself from the students and how this interaction is contingent on her self-analysis'.

## 5. Discussion

This qualitative, year-long AR study, conducted in the context of PETE, aimed to explore how PSTs deconstruct their initial reflections (i.e., their initial reflections about reflection concepts) and subsequently re-signify their understanding based on challenges encountered in real-world practicum settings. Moreover, it also examined how the facilitator supported the PSTs' reflections over time. Our study addresses a gap highlighted in Standal and Moe's systematic review, which asserts that PSTs in PE often lack the necessary knowledge for critical reflection. Indeed, the main value of our work lies in the ability to employ the EF's clarification and deconstruction of reflection concepts in a fully situated and contextually tailored manner, addressing each PST's distinctive experiences during their practicum. The EF skillfully guided this deconstruction by posing critical questions and challenging assumptions, prompting the PSTs to actively engage in the re-examination and questioning of their own practices.

This approach enabled the PSTs to recognize not only the limitations to their initial analysis of teaching–learning practices, but also to deepen their understanding of essential concepts of reflection. As a result, the PSTs were not mere recipients of knowledge, but actively participated in the deconstruction process, bringing their unique insights and experiences to the fore. Such renewed awareness led them to resignify their understanding of reflection, enriched by the interpretation of challenges that emerged in their teaching–learning practice. Through this collaborative and interactive process, the PSTs gained

renewed awareness, leading them to resignify their understanding of reflection. This understanding was further enriched by their interpretation of challenges that emerged in their teaching–learning practice.

Indeed, in our study, theoretical concepts proved to be essential for PSTs in understanding the meaning of reflection. However, it was the interpretation of real-life challenges faced by the PSTs during their practicum that acted as a catalyst for internalizing and redefining their understandings of reflection. When PSTs had the opportunity to analyze these theoretical concepts in light of their own experiences and realities, they not only appropriated these concepts, but were also able to create new understandings. This explicit process facilitated the optimized development of their growth as reflective educators, demonstrating the power of practice in enriching and shaping theory through the creation of tangible meaning. Our approach was guided by a theoretical framework that facilitated ongoing data analysis, establishing a strong link between various reflection concepts and the specific challenges that the PSTs encountered at different stages of their practicum.

During the first cycle of AR, the facilitator sought to gauge the PSTs' initial understanding of reflection and its application in their teaching–learning practice. The identification of the PSTs' initial understandings of the meaning of reflection was pivotal, as it provided a solid foundation that enabled the facilitator to select and adapt specific strategies for deconstructing and deepening their conceptual understanding and practical application of reflection [9]. Our findings reveal that PSTs understood reflection as a process of 'checking aspects,' such as remembering plans, counting students, and considering materials. This limited comprehension hinders their ability to become reflective practitioners who learn continuously from their experiences and reconstruct their understanding through reflection [24]. Our findings reinforce the perspective emphasized in the literature, which is that PSTs often focus on the superficial aspects of PE lessons [19], neglecting the deeper structure due to the demands of critical thinking and problem-solving processes [13]. The struggle among PSTs to fully understand reflection, along with their focus on instructional aspects in PE, underscores the urgent need for a more comprehensive explanation and deconstruction of what reflection truly involves.

Based on the understanding of the PSTs' limitations regarding what reflection meant to them, the facilitator aimed to help them become aware that their current way of reflecting was constricting. To address this, in the second AR cycle, the facilitator identified a crucial need to explicitly deconstruct and clarify, firstly, the concepts of reflection and reflective practice, and later, critical reflection. Through the deconstruction of the concepts in light of the unique challenges faced by the PSTs, they were able to recognize that they were not aware of what it meant to reflect—'wait, so we weren't reflecting, but just stating'. They also realized that understanding its meaning would help them teach effectively—'(. . .) in that way, it could help me improve my lessons'.

The EF's support helped the PSTs navigate the nuances of reflection and reflective practice (i.e., questioning, integrating new ideas and looking for new alternatives), ultimately enhancing their understanding and leading them to reframe their initial perspectives. This resignification was evident in their queries about the relevance of incorporating new elements into their reflections, such as considerations on 'thinking about other things,' 'getting more involved,' and 'taking responsibility for decision-making'. Instead of merely conforming, PSTs began to seek reasons for what occurred in their experiences: 'I know that in the next class since they don't like the subjects that day, I'll have to think of a class that they'll enjoy more (. . .)'. This initial questioning was not an isolated incident but became a sustained practice. As articulated in our theoretical framework, this suggests that PSTs were not merely questioning ideas in a sporadic manner, but were consistently exploring new alternatives. This sustained engagement indicates a genuine awareness and understanding of the reflection process. While the literature often emphasizes the importance of deconstructing and reconstructing experiences for effective reflection, it commonly overlooks the initial misconceptions that PSTs may have [14]. Our approach addresses this gap by focusing on the PSTs' initial understanding, thus providing a more rounded view

of how reflection and reflective practice concepts can be meaningfully internalized. Our findings suggest that gaining a robust initial understanding of the concept of reflection and critical reflection not only enabled the PSTs to truly value the concept, but also served as a strong foundation for them to reframe their thinking regarding their teaching–learning practices [41,42]. However, PSTs still seemed to struggle with incorporating additional elements of critical reflection, such as the consideration of others and the context, into their critical reflection process.

During the third AR cycle, the facilitator recognized that although the PSTs were aware of the challenges of critical reflection, they had not yet fully understood its significance for developing self-awareness (i.e., incorporating self, others and context when analyzing issues). Building on the PSTs' existing theoretical understanding of reflection, the facilitator turned the focus towards deconstructing their teaching–learning practice to illuminate how theory could be operationalized.

For instance, the facilitator encouraged peer observation to help PSTs realize the necessity of having multiple solutions for the same problem (i.e., incorporating others when analyzing issues). Through this exercise, PSTs grasped not only the importance of being prepared with various strategies, but also recognized that analyzing a situation from multiple perspectives is a form of critical reflection. Similarly, when the facilitator prompted PSTs to explain specific events previously documented in their reflective journals, the strategy proved to be effective in enhancing their understanding of the continuous nature of reflection (i.e., uncovering the ongoing nature of reflection). This was crucial for grasping an advanced level of awareness regarding the meaning of critical reflection. This not only enabled them to transition from a theoretical understanding, but also to develop their own personally situated meanings (i.e., formulating fresh, unique personal meanings), which were based on different problems specific to their teaching practice, and initiate their practical application. This advancement resonated with the existing literature, such as Liu (2015) [18] and Szenes and Tilakaratna [68], emphasizing self-knowledge and the transformative power of reflection. This skill development allowed them to construct practical frameworks for understanding their roles, highlighting the ongoing need for reflective practice for professional growth. Although recent studies have emphasized the importance of a situated perspective on reflection, the development of reflection among PSTs has often been limited to their portfolios, lacking individualized guidance or additional support [22]. In our study, the individualized support promoted the PSTs to incorporate self, others, and context when analyzing issues. They made deliberate choices based on reasoning and method: 'I don't blame everyone and everything now, I know that above all, I am the one steering the ship, so I assume my decisions (...)'.

Our study highlighted that reflection tends to be a blind spot for PSTs in PE, especially concerning their understanding of it. These findings illuminate the pressing necessity for an initial analysis and understanding of how PSTs actually comprehend what reflection means and how they apply it to their own teaching–learning experiences. Drawing on the PSTs' initial understanding, facilitated by the EF's ongoing, flexible, situated, and contextualized support, the EF engaged them in a process to deconstruct the meaning of reflective thinking. This approach enables PSTs to engage intimately in reflective activities directly linked to their experiences, aligning with the literature that advocates for the effectiveness of personally meaningful practice in grasping reflection [42]. The facilitator used practical examples to deconstruct theory, enhancing the PSTs' understanding of reflection. Conversely, practice was analyzed to illustrate how theoretical concepts could be applied. This bidirectional approach enriched the PSTs' ability to reflect by enabling them to reframe their initial understandings based on practical application.

Despite making progress in understanding the three different concepts of reflection, the PSTs continued to struggle with incorporating their own perspectives in problem analysis. This challenge highlights the necessity for the facilitator's work to be personalized, adapted, and individualized to meet their specific needs.

## 6. Final Thoughts

Across the three AR cycles, our findings underscore the significance of deconstructing reflection concepts through the meaningful experiences of PSTs, such as during their practicum. This approach catalyzed the reflection process, as the facilitator prompted PSTs to scrutinize their actions and decisions in authentic contexts. By addressing practical and relevant issues, the PSTs grasped the applicability of reflection concepts in their own experiences, imbuing them with tangible meaning.

In sum, our research highlights the importance of integrating reflection concepts into the practical and challenging situations PSTs face. By merging theoretical concepts with real-world practice, the PSTs internalized reflection concepts more profoundly and developed the ability to apply them critically and intentionally in their teaching–learning activities. However, our study also underscores the necessity for systemic changes in teacher education. The role of facilitating advanced levels of reflective practice should not be limited to external facilitators alone; this role should also be integrated into the responsibilities of cooperating teachers. To effectively assume this role, cooperating teachers can pursue various avenues for professional learning, including, but not limited to, specialized training, self-guided educational initiatives, mentorship from peers, collaboration with educational facilitators, and the employment of technological solutions for structured reflection.

**Author Contributions:** Conceptualization, E.A. and I.M.; methodology, E.A. and A.R. and C.V.; formal analysis, E.A. and I.M. and A.R.; investigation, E.A. and I.M.; writing—original draft preparation, E.A.; writing—review and editing, C.V. and R.A.; visualization, A.R.; supervision, I.M. and R.A.; project administration, I.M.; funding acquisition, E.A. All authors have read and agreed to the published version of the manuscript.

**Funding:** This research was funded by PORTUGUESE FOUNDATION FOR SCIENCE AND TECHNOLOGY (FCT), grant number SFRH/BD/134292/2017.

**Institutional Review Board Statement:** The study was conducted in accordance with the Declaration of Helsinki, and approved by the Ethics Committee of the Faculty of Sports from the University of Porto (CEFADE 29 2023).

**Informed Consent Statement:** Informed consent was obtained from all subjects involved in the study.

**Data Availability Statement:** Due to the confidential nature of the data, it was not possible to make the underlying data for this research available.

**Conflicts of Interest:** The authors declare no conflict of interest.

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
