# Peer review of "Diving into Real-World Practicum in Physical Education: Deconstructing and Re-Signifying Pre-Service Teachers’ Reflections"

_education, doi:10.3390/educsci14010011_

Round 1

Reviewer 1 Report

Comments and Suggestions for Authors

Abstract

l. 10: "AR" should be replaced by "Action research" the first time it occurs.

l. 16: What "these concepts" means is unclear. Does it refer to PSTs' views on reflection (l. 15)? To the three dimensions of reflection as developed in the theoretical framework?

Introduction

The introduction is well structured and leads quite logically to the two research questions.
Nevertheless, the first one involves the process of deconstruction-resignification, which is not developed in the introduction nor in the theoretical framework and is therefore open to different interpretations. Furthemore, what seems to be deconstructed and resignificated is their initial reflections on the concept of reflection. This particularity should be clarified as it leads to confusion.

Indeed, we can wonder why the content of the PSTs' reflections (i.e. what happened during planification and lessons) is not considered more in your analyses when you write: "However, the PSTs are often unaware of both their level of reflection and the specific topics they are considering. This requires a deliberate approach that includes both the content and process of reflection" (l. 55-56).

Theoretical framework

The theoretical framework is composite and rather rudimentary. On one hand, it does not explain why the three dimensions of reflection (reflection, reflective practice and critical reflection) are crucial for enabling PSTs to grasp the essence of reflection in their teaching-learning practice. Considering these dimensions as concepts creates confusion when it comes to concepts later in the text (e.g., l. 119, 158, 207, 490).

The use of the same term to refer to the concept of reflection and the first of its three dimensions also creates confusion. Furthermore, the first (reflection) and third (critical reflection) of these dimensions appear to be positioned in a hierarchical order. The results indicate that the FE seeks to move the PSTs towards critical reflection. The status of the second dimension (reflective practice) in this hierarchy is less clear .

On the other hand, the process of deconstruction-resignification, which appears repeatedly in the results, should be developed in the theoretical framework. As I understand it, this process refers to learning from a constructivist perspective, which consists of replacing false or insufficient knowledge (deconstruction) with new knowledge (resignification). However, the way in which it is used in the description of the results does not correspond to this meaning (see below).

Methodology

The methodology is clearly stated and coherent with your two research questions, except for the following point.

You write that "a qualitative approach was employed in this study to investigate the PSTs' understanding on reflection and the influence of reflection on PSTs' practicum" (l. 138-139). However, the second object of investigation is not studied. This would have required, as stated above, to be more interested in the contents of the reflection. Therefore, I suggest you to rephrase this sentence, reintroducing the second research question, i.e. how the facilitator supported the PSTs' reflections over time.

Results and Discussion

These sections are clearly presented, in a chronological order that makes sense.

In the results section, the numbers of the three AR-cycles should be separated from the numbers of the subsection titles.

The most problematic point of these sections concerns, as already mentioned, the use of the process of deconstruction-resignification. In the results section, the term "deconstruction" is almost exclusively used to characterize the action of the facilitator (ex. l. 340: "The deconstruction of concepts by the facilitator, through specific examples from the PSTs' practice..."). What does it mean to deconstruct the concepts of reflection for the facilitator? Would it not be more appropriate to say that the facilitator explains these concepts?

In the discussion section, the term "deconstruction" is used to characterize both PSTs and the facilitator's actions. But can we consider that these are the same actions? Furthermore, what is supposed to be deconstructed and resignificated is ambiguous: understanding of the concept(s) of reflection? Teaching-learning practice?

examples:

...How PSTs deconstruct their initial reflections and subsequently re-signify their understanding... (l. 478-479): here the object of deconstruction is unclear (their initial reflections about reflection concepts? Their teaching-learning practice?).

...the EF's clarification and deconstruction of reflection concepts (l. 483-484): here the object of deconstruction is clear (reflection concepts) but can it really be termed deconstruction?

...the literature often emphasizes the importance of deconstructing and reconstructing experiences for effective reflection (l. 537-538): here, the object of deconstruction seems clear (experiences of teaching) but what does it mean to deconstruct and reconstruct experiences for a learner?

...the facilitator turned the focus towards deconstructing their teaching-learning practice (l. 550-551): here the object of deconstruction is clear (teaching-learning practice) but what does it mean to deconstruct and reconstruct practice for a learner?

Author Response

Dear Revisor 1

I would like to express my sincere gratitude for the observations and questions raised regarding our manuscript. I am thankful for the detailed attention and constructive insights that were provided.

The issues highlighted  have been extremely helpful in enhancing our work and ensuring that the results and conclusions are presented in the most clear and robust manner possible and the suggesions have allowed us to reflect on and expand certain aspects of our research, significantly contributing to the quality of the manuscript.

Thank you once again for the opportunity to improve our work and for the valuable guidance provided.

Response to Reviewer

Manuscript: Diving into Real-World Practicum in Physical Education: Deconstructing and Re-signifying Pre-service Teachers' Reflections

ID: education-2711024

Point raised by Reviewer: 1

Author's Response

Observation

l. 10: "AR" should be replaced by "Action research" the first time it occurs.

 The change has been made as suggested, replacing "AR" with "Action Research" at the first occurrence in the text. (l. 10)

l. 16: What "these concepts" means is unclear. Does it refer to PSTs' views on reflection (l. 15)? To the three dimensions of reflection as developed in the theoretical framework?

We have revised the sentence to clearly specify that "these concepts" refer to the three concepts developed in our theoretical framework and how they are deconstructed through the examination of the PSTs' real teaching-learning experiences. (l. 16)

Introduction

The introduction is well structured and leads quite logically to the two research questions.
Nevertheless, the first one involves the process of deconstruction-resignification, which is not developed in the introduction nor in the theoretical framework and is therefore open to different interpretations. Furthermore, what seems to be deconstructed and resignificated is their initial reflections on the concept of reflection. This particularity should be clarified as it leads to confusion.

Indeed, we can wonder why the content of the PSTs' reflections (i.e. what happened during planification and lessons) is not considered more in your analyses when you write: "However, the PSTs are often unaware of both their level of reflection and the specific topics they are considering. This requires a deliberate approach that includes both the content and process of reflection" (l. 55-56).

We agree that in the previous version of the introduction, the deconstruction-resignification process was not explicitly outlined, which could have led to various interpretations. We appreciate the critical observation and, recognizing the importance of clarifying this aspect, have made a conscious effort to correct this oversight.

In response to your observation regarding the need to clarify the focus of deconstruction and resignification, we have revised the introduction to highlight that, although it may initially appear that we are deconstructing and resignifying only the PSTs' initial reflections on the concept of reflection, our focus is actually broader. We recognize that it is crucial to consider both the content and the process of the PSTs' reflection. Thus, we clarified in the introduction that our deliberate approach includes analyzing the content of the PSTs' reflections (e.g., their planning and teaching experiences) and understanding and nurturing the reflective process itself.

Additionally, as you suggested, we removed one of the research objectives related to studying the specific content of the PSTs' reflections during planning and lessons. Therefore, although the introduction considers the reflection process in a broad sense, the main focus is on how the PSTs comprehend and deconstruct their theoretical and conceptual understandings of reflection, and not on the specific details of what occurred during their planning and teaching activities. This aligns the introduction with the revised objectives of the study, maintaining consistency throughout the work.

(2. 54-69)

Theoretical framework

The theoretical framework is composite and rather rudimentary. On one hand, it does not explain why the three dimensions of reflection (reflection, reflective practice and critical reflection) are crucial for enabling PSTs to grasp the essence of reflection in their teaching-learning practice. Considering these dimensions as concepts creates confusion when it comes to concepts later in the text (e.g., l. 119, 158, 207, 490).

The use of the same term to refer to the concept of reflection and the first of its three dimensions also creates confusion. Furthermore, the first (reflection) and third (critical reflection) of these dimensions appear to be positioned in a hierarchical order. The results indicate that the FE seeks to move the PSTs towards critical reflection. The status of the second dimension (reflective practice) in this hierarchy is less clear .

On the other hand, the process of deconstruction-resignification, which appears repeatedly in the results, should be developed in the theoretical framework. As I understand it, this process refers to learning from a constructivist perspective, which consists of replacing false or insufficient knowledge (deconstruction) with new knowledge (resignification). However, the way in which it is used in the description of the results does not correspond to this meaning (see below).

Thank you for sharing with us your concern regarding the appearance of a hierarchical relationship between these dimensions and the potential confusion it might create and we totally agree that it requires further clarification. In response, we have revised the theoretical framework to clarify that, although these concepts can be conceptually interpreted as hierarchical, the decision to include them was driven by the analysis of the practical difficulties faced by PSTs. This approach highlights the interrelated and complementary nature of these dimensions, arising in direct response to the needs and challenges encountered by PSTs in their teaching experiences.

Furthermore, we have integrated a more detailed explanation of the deconstruction-resignification process into the theoretical framework. This process, as evidenced in our results, is interpreted as a constructivist approach to learning, where previous or insufficient knowledge (deconstruction) is replaced with new knowledge (resignification). We clarified how this process specifically applies in the context of the PSTs' teaching-learning experiences, emphasizing its vital role in facilitating deep understanding and transforming teaching practices.

(3. 123-159)

Methodology

The methodology is clearly stated and coherent with your two research questions, except for the following point.

You write that "a qualitative approach was employed in this study to investigate the PSTs' understanding on reflection and the influence of reflection on PSTs' practicum" (l. 138-139). However, the second object of investigation is not studied. This would have required, as stated above, to be more interested in the contents of the reflection. Therefore, I suggest you to rephrase this sentence, reintroducing the second research question, i.e. how the facilitator supported the PSTs' reflections over time.

Thank for your input in clarifying this section. Following your suggestion, we have revised the sentence in question to more accurately reflect the revised aims of our study. We wrote: ‘A qualitative approach was employed in this study to investigate not only the PSTs' understanding of reflection but also how the facilitator supported their reflections during the practicum.’. Indeed, the facilitator's work throughout the action research during the school year is evident in the results, helping PSTs to deconstruct and ressignified their understanding of reflection concepts.

(4. 161-162)

Results and Discussion

These sections are clearly presented, in a chronological order that makes sense.

In the results section, the numbers of the three AR-cycles should be separated from the numbers of the subsection titles.

The most problematic point of these sections concerns, as already mentioned, the use of the process of deconstruction-resignification. In the results section, the term "deconstruction" is almost exclusively used to characterize the action of the facilitator (ex. l. 340: "The deconstruction of concepts by the facilitator, through specific examples from the PSTs' practice..."). What does it mean to deconstruct the concepts of reflection for the facilitator? Would it not be more appropriate to say that the facilitator explains these concepts?

In the discussion section, the term "deconstruction" is used to characterize both PSTs and the facilitator's actions. But can we consider that these are the same actions? Furthermore, what is supposed to be deconstructed and resignificated is ambiguous: understanding of the concept(s) of reflection? Teaching-learning practice?

examples:

...How PSTs deconstruct their initial reflections and subsequently re-signify their understanding... (l. 478-479): here the object of deconstruction is unclear (their initial reflections about reflection concepts? Their teaching-learning practice?).

...the EF's clarification and deconstruction of reflection concepts (l. 483-484): here the object of deconstruction is clear (reflection concepts) but can it really be termed deconstruction?

...the literature often emphasizes the importance of deconstructing and reconstructing experiences for effective reflection (l. 537-538): here, the object of deconstruction seems clear (experiences of teaching) but what does it mean to deconstruct and reconstruct experiences for a learner?

...the facilitator turned the focus towards deconstructing their teaching-learning practice (l. 550-551): here the object of deconstruction is clear (teaching-learning practice) but what does it mean to deconstruct and reconstruct practice for a learner?

We appreciate your concerns regarding the distinction between "explaining" and "deconstructing" the concepts of reflection. However, we believe that the term "deconstruction" is more appropriate in this context than merely "explaining."

The act of "explaining" the concepts of reflection would imply transmitting information or knowledge directly, focusing on clarifying and presenting the concepts as they are traditionally or establishedly understood. In contrast, "deconstructing" involves a deeper and more interactive process, in which the facilitator assists the PSTs in questioning, critically examining, and eventually rethinking their prior understandings of the concepts of reflection. This process is not limited to providing information; it challenges existing notions, promoting a re-evaluation and possibly a transformation in how the PSTs understand and apply these concepts in their practice. To add further clarification, we added: “Deconstruction refers to critically examining and questioning existing practices and understandings, particularly in response to challenges encountered in their practicum”.

In this context, "deconstruction" refers to the questioning and critical analysis of situations and concepts, as they are situated in practice. The facilitator begins by deconstructing concepts and situations, questioning and provoking reflection. This approach is not merely explanatory, but rather a support for the PSTs to, in turn, think critically and embark on their own journey of deconstruction and subsequent resignification. This is a dual interplay process: initially led by the facilitator and then by the PSTs, who develop the ability to dismantle and reinterpret the concepts for themselves.

Incorporating this deconstruction-resignification process is essential for a constructivist approach in teaching and learning. By viewing the facilitator as someone who supports the deconstruction process, we are effectively including a constructivist practice. This process not only helps the PSTs to understand the concepts more deeply but also empowers them to apply them in a critical and reflective manner in their own practices.

Therefore, the choice of words in the results section was intentional to capture the complexity and depth of the facilitated learning process, which goes beyond the mere explanation of concepts, engaging the PSTs in an active process of deconstruction and resignification.

Based on your comments, we deemed it appropriate and necessary to clarify and detail the process of deconstruction and resignification in the study's methodology. This clarification now includes a more in-depth description of the facilitator’s role in guiding the PSTs through this process, as well as an explanation of how the PSTs actively engage in the deconstruction of their own practices and initial reflections. Furthermore, we have detailed the specific objects of deconstruction and resignification, whether in terms of reflection concepts or teaching-learning practices.

We believe that this complementary clarification in the methodology will provide a solid foundation for the interpretation of the results and the discussion, illuminating any ambiguities and addressing the concerns raised. (6. 255-270)

Moreover, recognizing the importance of this clarity, we also made a concerted effort to further elucidate this process in the discussion section. This dual approach ensures that both the methodology and the discussion comprehensively address and illuminate the complex dynamics of the facilitator's role and the active participation of the PSTs in the deconstruction and resignification process.

(11. 521-533)

Reviewer 2 Report

Comments and Suggestions for Authors

The article presents a high level of scientific quality.
This fact must be praised.
Its originality, its contribution to research in the scientific field, its methodological precision, the clarity of its written exposition and the depth of its reflections should be highlighted.

  1. What is the main question addressed by the research? 

The aim of the study was to explore how PSTs deconstruct their initial reflections and subsequently re-signify their understanding, based on challenges encountered in real-world practicum settings. 

2. Do you consider the topic original or relevant in the field? Does it
address a specific gap in the field? 

Yes, because there remains a noticeable gap in the literature regarding how PSTs effectively internalize the concepts and processes of reflection, as well as the difficulties they encounter in recognizing the personal effort that reflection demands. 

3. What does it add to the subject area compared with other published
material? 

The research highlights the importance of integrating reflection concepts into 

practical and challenging situations PSTs face. By merging theoretical concepts with real-world practice, PSTs internalized reflection concepts more profoundly and developed the ability to apply them critically and intentionally in their teaching-learning activities. However, this study also underscores the necessity for systemic changes in teacher education. 

4. What specific improvements should the authors consider regarding the
methodology? What further controls should be considered? 

The methodology is presented clearly, with depth and scientific rigor. 

5. Are the conclusions consistent with the evidence and arguments presented
and do they address the main question posed? 

Yes, they are consistent. 

6. Are the references appropriate? 

Yes, the article is supported by a rich and updated bibliography. 

7. Please include any additional comments on the tables and figures. 

No comments.

Author Response

Dear Revisor,

Your recognition of the scientific quality, originality, and methodological precision of our work is immensely gratifying. We are particularly thankful for your acknowledgment of the clarity of our exposition and the depth of our reflections.

We are pleased to inform you that we have diligently addressed all the points and suggestions you raised. In our revised manuscript, we have made concerted efforts to enhance these aspects further, ensuring that our contribution to the scientific field is as impactful and clear as possible.

Your positive feedback and constructive guidance have been instrumental in refining our work. We believe that the revisions made not only align with your recommendations but also significantly bolster the overall strength and coherence of the manuscript.

We have attached the revised version of the manuscript for your review. We hope that our revisions meet your expectations and further the scholarly discourse in our field.

Thank you once again for your valuable contribution to our work. We look forward to any further suggestions you might have and to the possibility of our manuscript making a significant contribution to the scientific community.

Best regards,

Round 2

Reviewer 1 Report

Comments and Suggestions for Authors

You took all my comments into account and significantly improved the manuscript. The confusions that I had pointed out in my first review have been clarified.